# Development of a nucleoside-modified mRNA vaccine against clade 2.3.4.4b H5 highly pathogenic avian influenza virus

Colleen Furey[1], Gabrielle Scher [1], Naiqing Ye [1], Lisa Kercher [2], Jennifer DeBeauchamp[2], Jeri Carol Crumpton[2], Trushar Jeevan[2], Christopher Patton [2,3], John Franks[2], Adam Rubrum[2], Mohamad-Gabriel Alameh [4,5], Steven H. Y. Fan[6], Anthony T. Phan[7], Christopher A. Hunter [7], Richard J. Webby [2], Drew Weissman[4,5] & Scott E. Hensley [1] ✉

mRNA lipid nanoparticle (LNP) vaccines would be useful during an influenza virus pandemic since they can be produced rapidly and do not require the generation of egg-adapted vaccine seed stocks. Highly pathogenic avian influenza viruses from H5 clade 2.3.4.4b are circulating at unprecedently high levels in wild and domestic birds and have the potential to adapt to humans. Here, we generate an mRNA lipid nanoparticle (LNP) vaccine encoding the hemagglutinin (HA) glycoprotein from a clade 2.3.4.4b H5 isolate. The H5 mRNA-LNP vaccine elicits strong T cell and antibody responses in female mice, including neutralizing antibodies and broadly-reactive anti-HA stalk antibodies. The H5 mRNA-LNP vaccine elicits antibodies at similar levels compared to whole inactivated vaccines in female mice with and without prior H1N1 exposures. Finally, we find that the H5 mRNA-LNP vaccine is immunogenic in male ferrets and prevents morbidity and mortality of animals following 2.3.4.4b H5N1 challenge. Together, our data demonstrate that a monovalent mRNA-LNP vaccine expressing 2.3.4.4b H5 is immunogenic and protective in pre-clinical animal models.

Highly pathogenic avian influenza (HPAI) H5 viruses of the A/goose/Guangdong/1996 (Gs/Gd) lineage emerged in southeast Asia in 1996 and have since spread geographically and diversified into several genetically distinct hemagglutinin (HA) clades[1–3]. Long-distance migration of wild birds has enabled the rapid transcontinental spread of these HPAI viruses, as evidenced by past H5 outbreaks in 2005-2006, 2014-2015, and 2016-2017[3–5]. Upon re-emerging in 2020, Gs/Gd lineage H5 viruses of clade 2.3.4.4b have circulated at historically high levels in wild and domestic bird populations across Europe, Asia, the Middle East, Africa, and North and South America[2,6–8].

2.3.4.4b H5 viruses have persisted with outbreaks continuing uncharacteristically over the summer seasons, wreaking havoc on the poultry industry and resulting in high rates of wild bird mortality[6,9–11]. In comparison to previous H5 outbreaks, a wider range of wild and domestic bird species have been affected by the spread of clade

[1]Department of Microbiology, Perelman School of Medicine, University of Pennsylvania, Philadelphia, PA, USA. [2]Department of Host-Microbe Interactions, St. Jude Children's Research Hospital, Memphis, TN, USA. [3]Department of Microbiology, Immunology, and Biochemistry, University of Tennessee Health Science Center, Memphis, TN, USA. [4]Infectious Disease Division, Perelman School of Medicine, University of Pennsylvania, Philadelphia, PA, USA. [5]Penn Institute for RNA Innovation, Perelman School of Medicine, University of Pennsylvania, Philadelphia, PA, USA. [6]Acuitas Therapeutics, Vancouver, BC, Canada. [7]Department of Pathobiology, School of Veterinary Medicine, University of Pennsylvania, Philadelphia, PA, USA. ✉e-mail: hensley@pennmedicine.upenn.edu

2.3.4.4b H5 viruses since 2020[12,13]. There have also been occasional human infections and increasing incidences of clade 2.3.4.4b H5 virus spillover into mammals such as cows, red foxes, seals, and minks[14–17]. Some viruses isolated from infected mammals contain genetic mutations associated with mammalian adaptation, highlighting the potential risk an expanded host range can pose[13,15,17–20].

Our laboratory and others previously demonstrated that mRNA-lipid nanoparticle (LNP) vaccines encoding influenza virus HA induce potent immune responses in mice, rabbits, and ferrets, and clinical trials confirm their safety and immunogenicity in humans[21–24]. We recently developed a multivalent mRNA-LNP vaccine that encodes an HA protein from every influenza virus subtype, including clade 1 H5[24]. This multivalent vaccine protects experimentally infected animals against severe disease and death when challenge strains are antigenically mismatched to the vaccine immunogens[24]; however, the vaccine is not expected to elicit neutralizing antibodies and sterilizing immunity against mismatched influenza virus strains, such as clade 2.3.4.4b H5 viruses. It is therefore important to also develop tailored-made vaccines precisely matched to influenza virus strains with high pandemic potential.

Here, we create a monovalent mRNA-LNP encoding HA from a 2.3.4.4b virus and we test this vaccine in mice and ferrets. We show that the vaccine is immunogenic in mice with and without prior H1N1 exposures. Mice that receive the H5 mRNA-LNP vaccine produce antibodies that efficiently neutralize the 2.3.4.4b H5 virus, as well as antibodies that can bind to the HA stalk of diverse H5 viruses. We show that the vaccine is also immunogenic in ferrets and prevents morbidity and mortality of ferrets following the 2.3.4.4b H5N1 challenge.

## Results

### 2.3.4.4b HA mRNA-LNP is immunogenic in mice
We created a monovalent nucleoside-modified mRNA-LNP vaccine encoding HA from the clade 2.3.4.4b A/Astrakhan/3212/2020 virus. We vaccinated female mice with 1 or 10 µg of H5 mRNA-LNP vaccine or 10 µg of a control mRNA-LNP vaccine expressing an irrelevant protein (Ovalbumin), and quantified serum antibody levels using ELISAs and neutralization assays. Both doses of the H5 mRNA-LNP vaccine elicited high levels of antibodies that bound to the full-length HA protein of A/Astrakhan/3212/2020 (Fig. 1A) and a 'headless' H5 stalk protein (Fig. 1B). Both doses of the vaccine also elicited antibodies that neutralized virus expressing the A/Astrakhan/3212/2020 HA (Fig. 1C). Serum antibody titers remained at high levels 1 year after vaccination (Fig. 1A–C).

We also tested antibody binding and neutralization of two additional 2.3.4.4b H5 strains, including A/red fox/England/AVP-M1-21-01/2020, which is an H5N8 strain isolated from a red fox, and A/pheasant/New York/22-009066-001/2022 which is an H5N1 virus representative of 2.3.4.4b strains currently circulating in the United States. Relative to the A/Astrakhan/3212/2020 H5 vaccine immunogen, the A/red fox/England/AVP-M1-21-01/2020 HA possesses P152S and S336N substitutions, and the A/pheasant/New York/22-009066-001/2022 HA possesses L120M, V226A, and I526V HA substitutions. The A/Astrakhan-3212/2020-based H5 mRNA-LNP vaccine elicited antibodies that bound (Fig. 2A, B) and neutralized (Fig. 2C, D) both A/red fox/England/AVP-M1-21-01/2020 and A/pheasant/New York/22-009066-001/2022. Antibody titers against these variant viruses were ~3-fold lower compared to A/Astrakhan/3212/2020 titers (Supplementary Fig. 1).

Since we found that the vaccine elicits high levels of antibodies that bind to the H5 stalk (Fig. 1B), we completed additional experiments to determine if these antibodies could bind to diverse H5 proteins via multianalyte bead-based experiments. Consistent with our initial experiments, we found that both doses of vaccine elicited antibodies that could bind to beads coupled with the 2.3.4.4b HAs from A/Astrakhan/3212/2020 (Fig. 3A), A/red fox/England/AVP-M1-21-01/2020 (Fig. 3B), and A/pheasant/New York/22-009066-001/2022 (Fig. 3C). The vaccine also elicited antibodies that bound to beads coupled with the HAs of a clade 1 virus (A/Vietnam/1203/2004; Fig. 3D), a clade 2.3.2.1a virus (A/Hubei/1/2010; Fig. 3E), and a clade 2.1.3.2 virus (A/Indonesia/5/2005; Fig. 3F), albeit binding was at reduced levels compared to binding to HAs from clade 2.3.4.4b viruses.

We also examined splenic T cell responses. For these experiments, we analyzed cells 10 days after vaccinating mice with 1 µg of H5 mRNA-LNP vaccine or 1 µg of a control mRNA-LNP vaccine. We found that the H5 mRNA-LNP vaccine elicited HA-specific CD8+ T cell responses (Fig. 4A, B). The H5 mRNA-LNP vaccine elicited polyfunctional CD8+

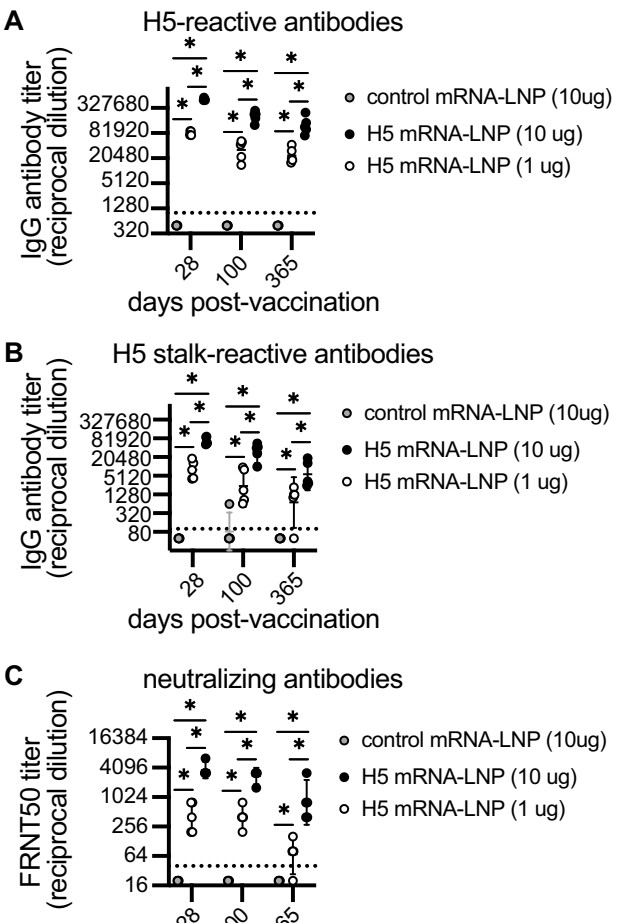

**Fig. 1 | Clade 2.3.4.4b H5 HA mRNA-LNP vaccine elicits long-lasting antibody responses in mice.** Five mice were included per experimental group. Mice were vaccinated i.m. with 1 or 10 µg A/Astrakhan/3212/2020 HA mRNA-LNP (H5 mRNA-LNP) or 10 µg Ovalbumin mRNA-LNP (control mRNA-LNP). Serum samples were collected from mice at 28, 100, and 365 days after vaccination and serum IgG reactive to the A/Astrakhan/3212/2020 recombinant full-length HA protein (**A**) or 'headless' H5 stalk protein (**B**) were quantified by ELISA. **C** A/Astrakhan/3212/2020 neutralizing antibodies were also quantified by a 50% Foci reduction neutralization test (FRNT50); reciprocal dilutions of serum required to inhibit 50% virus infection are shown. All data are shown as geometric means ± 95% confidence intervals. Data were compared using two-way ANOVA with Tukey's multiple comparisons test. Values were log-transformed before statistical analysis. Data are representative of 2 independent experiments. **A** all *$P < 0.0001$; (**B**) Day 365 control vs. 1 µg H5 mRNA-LNP *$P = 0.0009$, all others *$P < 0.0001$; (**C**) Day 28 1 µg H5 mRNA-LNP vs. 10 µg H5 mRNA-LNP *$P = 0.0015$, day 100 1 µg H5 mRNA-LNP vs. 10 µg H5 mRNA-LNP *$P = 0.0002$, day 365 1 µg H5 mRNA-LNP vs. 10 µg H5 mRNA-LNP *$P = 0.0018$, all other comparisons *$P < 0.0001$. Source data are provided as a Source Data file.

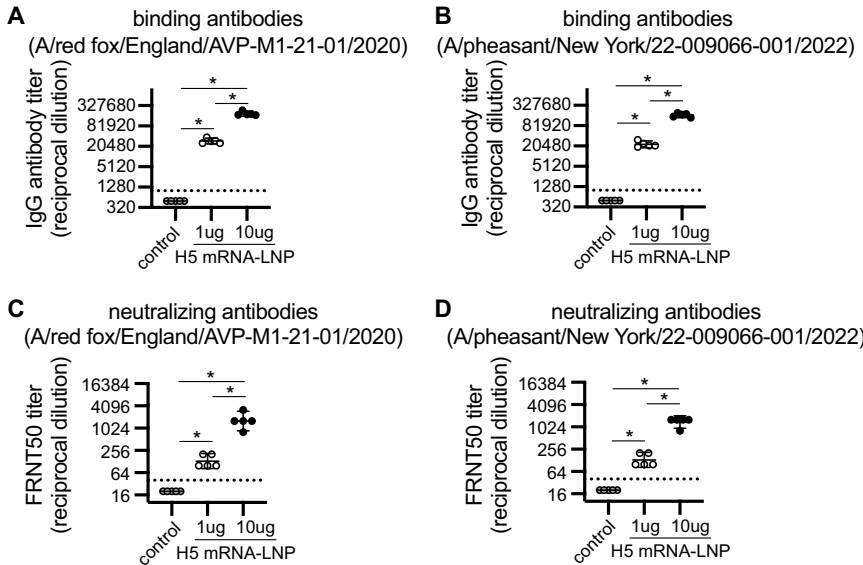

**Fig. 2 | Clade 2.3.4.4b H5 HA mRNA-LNP vaccine elicits antibodies that bind and neutralize different clade 2.3.4.4b H5 viruses.** Five mice were included per experimental group. Mice were vaccinated i.m. with 1 or 10 μg A/Astrakhan/3212/2020 HA mRNA-LNP (H5 mRNA-LNP) or 10 μg Ovalbumin mRNA-LNP (control). **A**, **B** Serum samples collected 28 days after vaccination were tested by ELISA to quantify IgG reactive to A/red fox/England/AVP-M1-21-01/2020 or A/pheasant/New York/22-009066-001/2022 recombinant HA proteins. **C**, **D** A/red fox/England/AVP-M1-21-01/2020 and A/pheasant/New York/22-009066-001/2022 neutralizing antibodies were quantified by 50% Foci reduction neutralization test (FRNT50) using serum samples collected 28 days after vaccination; reciprocal dilution of serum amounts required to inhibit 50% virus infection are shown. All data are shown as geometric means ± 95% confidence intervals. Data were compared using one-way ANOVA with Tukey's post hoc test. Values were log-transformed before statistical analysis. Data are representative of 2 independent experiments. *$P < 0.0001$. Source data are provided as a Source Data file.

CD44$^{high}$ CD62L$^{lo}$ T cells expressing Th1-associated cytokines IFNγ (Fig. 4C) and TNFα (Fig. 4D).

## Comparison of mRNA-LNP and inactivated vaccines in prior-exposure mouse model

Next, we completed an experiment to compare the monovalent H5 mRNA-LNP vaccine to a traditional β-propiolactone (BPL) inactivated whole virion vaccine. Since most humans have pre-existing immunity to seasonal influenza virus strains, we tested both vaccine types in mice with and without previous exposure to the A/California/07/2009 H1N1 virus (Fig. 5A). For these experiments, we immunized mice with either 1 μg of H5 mRNA-LNP vaccine or 50 hemagglutinating units (HAU) of an inactivated H5 vaccine. As expected, mice previously exposed to H1N1 possessed antibodies that bound to (Fig. 5B) and neutralized (Fig. 5C) A/California/07/2009, and these antibodies were not detected in mice without prior H1N1 exposure. The monovalent H5 mRNA-LNP vaccine elicited higher levels of antibodies that bound to the A/Astrakhan/3212/2020 HA compared to the inactivated vaccine, and antibody levels were similar in mice with and without prior H1N1 exposure (Fig. 5D). The two different vaccines elicited similar levels of A/Astrakhan/3212/2020 neutralizing antibodies in mice with and without prior H1N1 exposure (Fig. 5E). Taken together, these data suggest that both the monovalent H5 mRNA-LNP and inactivated H5 vaccine elicit robust antibody responses in mice previously exposed to H1N1 virus.

## 2.3.4.4b HA mRNA-LNP is immunogenic and protective in ferrets

We vaccinated male ferrets using a prime/boost strategy to mimic the dosing schedule initially used for severe acute respiratory syndrome coronavirus 2 (SARS-CoV-2) mRNA vaccination of humans[25,26]. Animals were primed with 60 μg of mRNA-LNP vaccine encoding H5 or an irrelevant protein (Luciferase) and then boosted 28 days later with the same vaccine. Ferrets vaccinated with H5 mRNA-LNP produced high levels of antibodies that bound and neutralized both the A/Astrakhan/3212/2020 (Fig. 6A, B) and A/pheasant/New York/22-009066-001/

2022 (Fig. 6C, D) viruses. The second dose of H5 mRNA-LNP boosted H5-reactive antibody levels ~8-fold higher relative to before the boost. Antibody titers against A/Astrakhan/3212/2020 and A/pheasant/New York/22-009066-001/2022 were similar (Supplementary Fig. 2). Similar to our murine experiments, we found that the vaccine elicited antibodies that bound to a 'headless' H5 stalk (Fig. 6E, F).

The vaccinated ferrets were then challenged with A/bald eagle/Florida/W22-134-OP/2022, an H5N1 strain that is similar to A/pheasant/New York/22-009066-001/2022 and previously shown to be lethal in ferrets[27]. The HAs of the A/Astrakhan/3212/2020 vaccine strain and the A/bald eagle/Florida/W22-134-OP/2022 challenge strain differ by 4 amino acids. All four H5 mRNA-LNP-vaccinated ferrets survived the H5N1 challenge, while all Luciferase mRNA-LNP-vaccinated animals reached clinical endpoint by 7 days post-challenge (Fig. 7A). Luciferase mRNA-LNP-vaccinated animals lost more weight (Fig. 7B), had detectable levels of virus in nasal washes for longer amounts of time (Fig. 7C), and displayed more clinical signs (Fig. 7D) relative to H5 mRNA-LNP-vaccinated animals.

## Discussion

We created a monovalent clade 2.3.4.4b H5 mRNA-LNP vaccine and demonstrated that it is immunogenic and protective in mice and ferrets. We found that the vaccine elicits robust antibody and CD8[+] T cell responses in mice, and we detected high levels of H5 antibodies 1 year after a single vaccination. The vaccine elicited neutralizing antibodies and broadly binding HA stalk antibodies in both mice and ferrets. In challenge experiments, we detected the virus in nasal washes obtained from vaccinated ferrets but these animals cleared the virus more rapidly than unvaccinated controls. Vaccinated ferrets lost less weight and displayed less clinical symptoms compared to unvaccinated animals after H5N1 challenge. All of the vaccinated animals in our study survived, whereas all of the unvaccinated animals died following the H5N1 challenge.

We found that the monovalent clade 2.3.4.4b H5 mRNA-LNP and an inactivated H5 vaccine both elicited high levels of antibodies in

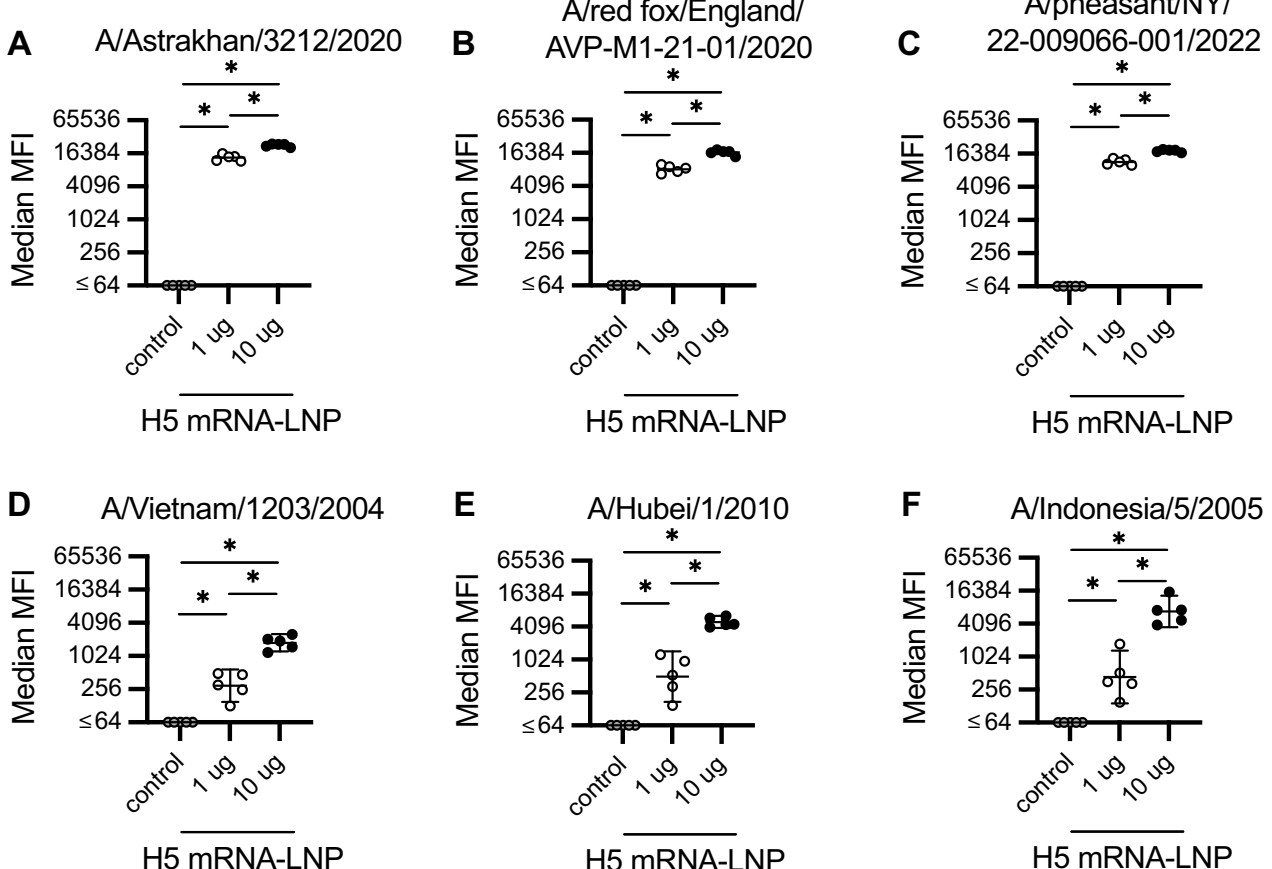

**Fig. 3 | Clade 2.3.4.4b H5 HA mRNA-LNP vaccine elicits antibodies that bind to diverse H5 viruses.** Groups of five mice were vaccinated i.m. with 1 or 10 μg A/Astrakhan/3212/2020 HA mRNA-LNP (H5 mRNA-LNP) or 10 μg Ovalbumin mRNA-LNP (control). Serum samples were collected 28 days after vaccination and serum antibody binding to HA proteins from diverse of H5 viral strains was determined using a bead-based multiplex binding assay. We measured antibody binding to the HA of (**A**) A/Astrakhan/3212/2020, (**B**) A/red fox/England/AVP-M1-21-01/2020, (**C**) A/pheasant/NY/22-009066-001/2022, (**D**) A/Vietnam/1203/2004, (**E**) A/Hubei/1/2010, and (**F**) A/Indonesia/5/2005. Binding is reported as mean fluorescence intensity (MFI) values. All data are shown as geometric means ± 95% confidence intervals. Data were compared using two-way ANOVA with Tukey's multiple comparisons test. Values were log-transformed before statistical analysis. Data are representative of 2 independent experiments. **A** 1 μg vs. 10 μg *$P = 0.004$; (**B**) 1 μg vs. 10 μg *$P = 0.0065$; (**E**) 1 μg vs. 10 μg *$P = 0.0002$; all other comparisons *$P < 0.0001$. Source data are provided as a Source Data file.

mice. It is difficult to directly compare these two different vaccine platforms, since it is unclear how vaccine doses in mice relate to vaccine doses in humans. We used a relatively low dose of mRNA-LNP vaccine (1 μg) and a relatively high dose of inactivated vaccine (50 HAU) for these comparative studies. Our experiments demonstrated that both vaccines elicited similar amounts of binding antibodies and no significant differences in neutralizing antibodies regardless of prior H1N1 exposures. A phase 1 study demonstrated that inactivated clade 2.3.4.4c H5N8 was well tolerated in humans and immunogenicity improved with higher vaccine amounts and multiple doses of adjuvanted vaccine[28]. It will be important to compare the clade 2.3.4.4b H5 mRNA-LNP vaccine to other vaccine platforms. A recent study demonstrated that a parainfluenza-based H5 vaccine delivered intranasally elicited antibodies that recognized diverse H5 proteins and protected animals against heterologous viral challenge, potentially through the induction of mucosal IgA responses[29]. A separate study demonstrated that recombinant A/Astrakhan/321/2020 HA proteins and propiolactone inactivated A/Astrakhan/321/2020 virus were both immunogenic in mice[30].

Our study has some limitations. We did not complete experiments to determine if the vaccine elicits antibodies that could neutralize H5 viruses from different clades. We found that the vaccine elicits antibodies that bind to the HAs of A/Vietnam/1203/2004, A/

Hubei/1/2010, and A/Indonesia/5/2005; however, it is likely that the observed cross-clade HA binding is mediated by non-neutralizing HA stalk antibodies. Another limitation is that we did not comprehensively test the H5 mRNA-LNP vaccine in animals with different prior exposures. We found that the vaccine elicits robust responses in mice previously exposed to H1N1 and additional studies should test how different prior exposures in mice and ferrets impact immunogenicity of the vaccine.

There are many potential benefits of mRNA-LNP based vaccines compared to inactivated vaccines. For example, during the 2009 H1N1 pandemic, conventional monovalent vaccines were distributed but there were manufacturing difficulties and these vaccines were not widely available until after the initial pandemic H1N1 viral waves subsided[31]. The capacity to make mRNA-LNPs has expanded widely since the start of the COVID-19 pandemic, and rapid production of mRNA-LNP vaccines is now possible. Notably, mRNA-LNP vaccines can be rapidly produced without first isolating and adapting viral strains that grow efficiently in fertilized chicken eggs or cell culture. It is now possible to start creating novel mRNA-LNP vaccines within hours of sequencing a new pandemic viral strain.

It will be important to evaluate the clade 2.3.4.4b H5 mRNA-LNP vaccine in birds and other animals that could potentially serve as an intermediate host to humans, such as swine and cattle. The recent

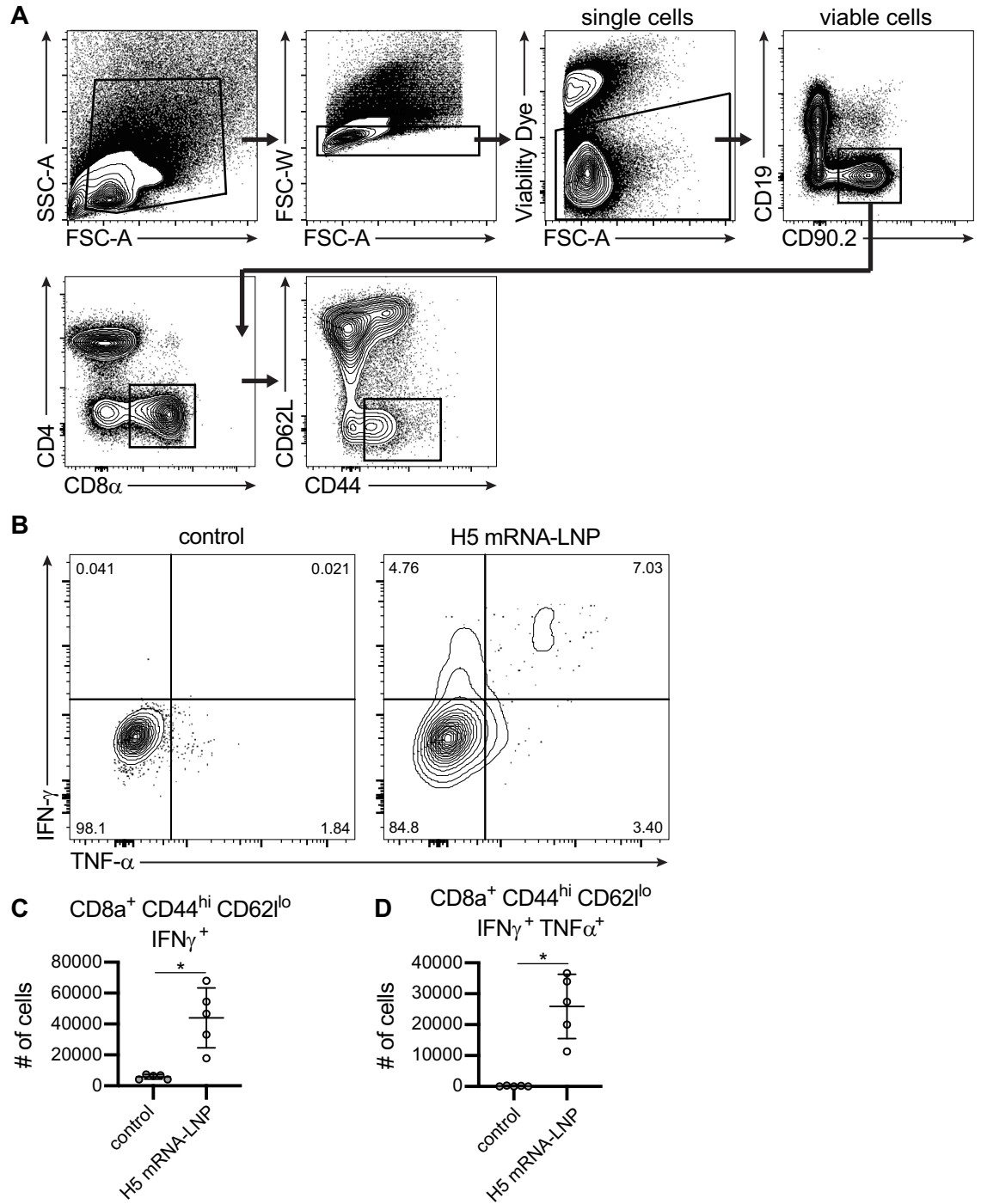

**Fig. 4 | Clade 2.3.4.4b H5 HA mRNA-LNP vaccine elicits robust CD8+ T cell responses.** Five mice per group were vaccinated i.m. with 1 μg A/Astrakhan/3212/2020 HA mRNA-LNP (H5 mRNA-LNP) or 1 μg Ovalbumin mRNA-LNP (control). Spleens were harvested at 10 days after vaccination and splenocytes were incubated with H5 HA overlapping peptide pools before completing intracellular cytokine staining and flow cytometric analysis of CD8 T cells. **A** Representative gating strategy for analysis of cytokine production by activated CD8+ T cells 10 days after vaccination. **B** Representative flow cytometry analysis of IFN-γ and TNF-α producing CD8+ T cells from mice vaccinated with control mRNA-LNP or H5 HA mRNA-LNP. **C**, **D** Number of IFN-γ producing CD8+ T cells and IFN-γ and TNF-α producing CD8+ cells were quantified. Data were compared using an unpaired two-tailed *t* test. Data are representative of 2 independent experiments. **C** *P = 0.0023; (**D**) *P = 0.0005. Source data are provided as a Source Data file.

2.3.4.4b H5 bovine outbreak is particularly alarming, due to the high number of apparent infections and potential human exposures. Our studies highlight the flexibility of the mRNA vaccine platform which allows for rapid development and precise antigenic matching of vaccine immunogens to emerging influenza virus strains with pandemic potential.

## Methods

### mRNA-LNP production

The HA sequence of A/Astrakhan/3212/2020 (H5) was codon-optimized, gene-synthesized by GenScript, and cloned into an mRNA production vector. mRNAs were then produced as previously described[32] using T7 RNA polymerase (Megascript, Ambion) on

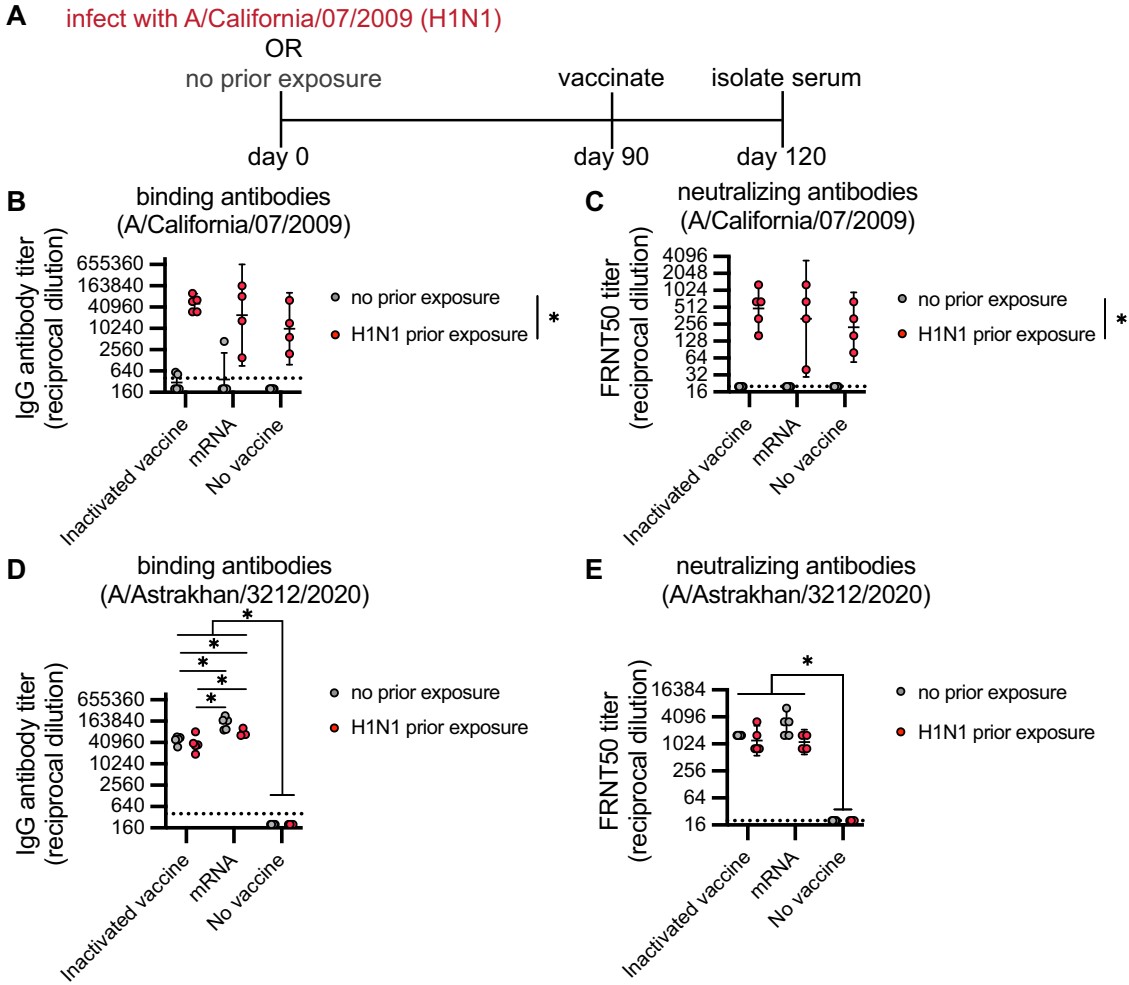

**Fig. 5 | Clade 2.3.4.4b H5 HA mRNA-LNP vaccine and inactivated H5 vaccine elicit robust antibody responses in mice previously exposed to H1N1 virus.**
**A** Timeline for infections and vaccinations. Groups of 4-5 mice were uninfected or infected i.n. with 1000 TCID$_{50}$ of A/California/07/2009 virus and then vaccinated i.m. 90 days later with either 1 μg of A/Astrakhan/3212/2020 HA mRNA-LNP (mRNA) or 50 HAU of an inactivated A/Astrakhan/3212/2020 vaccine (inactivated vaccine). Sera were isolated 30 days after vaccination and IgG antibody titers were measured using ELISA plates coated with HA from A/California/07/2009 (**B**) and A/Astrakhan/3212/2020 (**D**). 50% Foci reduction neutralization test (FRNT$_{50}$) titers were determined using viruses expressing the HAs from A/California/07/2009 (**C**) and A/Astrakhan/3212/2020 (**E**). For the 'no vaccine/no prior exposure' group, we tested

serum obtained from mice before initiating the experiment. All data are shown as geometric means ± 95% confidence intervals. Data were compared using two-way ANOVA with Tukey's multiple comparisons test. Values were log-transformed before statistical analysis. Data are representative of 2 independent experiments. **B** Inactivated vaccine no prior exposure vs. no vaccine prior exposure *$P = 0.0006$; mRNA no prior exposure vs. no vaccine prior exposure *$P = 0.0002$; (**C**) Inactivated vaccine no prior exposure vs. no vaccine prior exposure *$P = 0.0001$; (**D**) Inactivated vaccine no prior exposure vs. no vaccine no prior exposure *$P = 0.0444$; all other comparisons *$P < 0.0001$. Red indicates mice with prior exposure. Source data are provided as a Source Data file.

linearized plasmids. mRNAs were transcribed to contain 101 nucleotide-long poly(A) tails. Modified nucleoside-containing mRNA was generated using m1Ψ-UTP (TriLink) instead of UTP. In vitro, transcribed mRNAs were co-transcriptionally capped using the CleanCap (TriLink), and the mRNA purified using cellulose chromatography, as previously described[33]. All mRNAs were analyzed using native agarose gel electrophoresis, and tested to dsRNA and endotoxin content using dot blot and the LAL chromogenic assay respectively before storage at −20 °C.

Cellulose-purified, nucleoside-modified mRNAs were encapsulated in LNP using a self-assembly process as previously described[34] in which an ethanolic lipid mixture was rapidly mixed with an aqueous solution containing mRNA at pH = 4.0. The LNP used here contains an ionizable cationic lipid, phosphatidylcholine, cholesterol, and polyethylene glycol-lipid. The LNP has a mean hydrodynamic diameter of

-80 nm with a polydispersity index of 0.02–0.06 and an encapsulation efficiency of ~95%. The assembled mRNA-LNP were stored at −80 °C at a concentration of 1 μg/μL.

## Mouse experiments
All murine experiments were approved by the Institutional Animal Care and Use Committees of the Wistar Institute and the University of Pennsylvania. Mice were housed in ventilated, ducted Innovive racks with a 12- h light/dark cycle. The temperature and humidity are maintained according to *The Guide for the Care and Use of Laboratory Animals (NRC)*, with temperatures averaging 70–72 F and percent humidity averaging in 30's. Female C57BL/6 mice (Charles River Laboratories) aged 6–8 weeks were immunized intramuscularly with 1-10 μg of mRNA-LNP vaccine encoding either H5 or ovalbumin. For immunizations, mRNA-LNPs were diluted in 50 μl

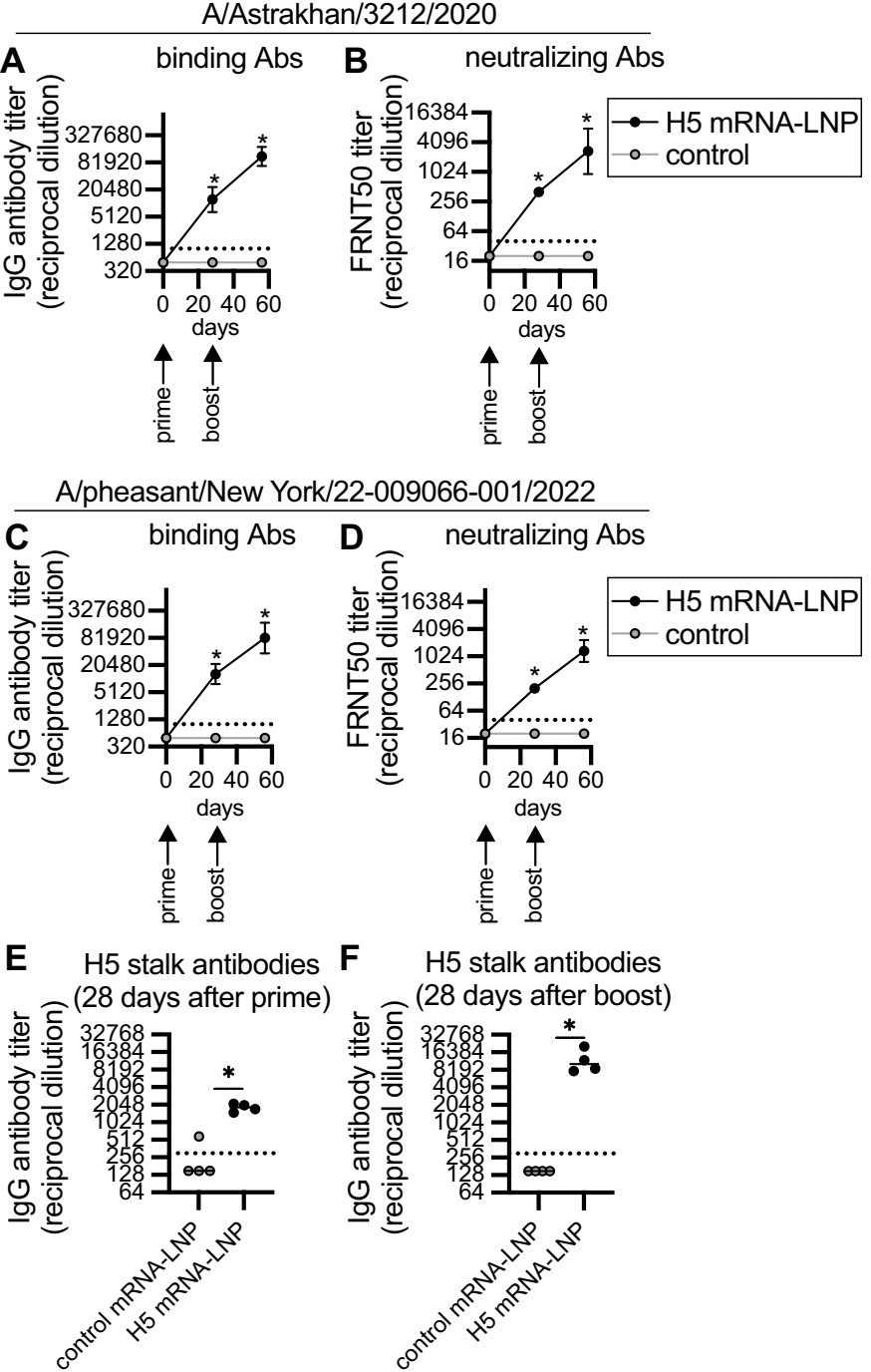

**Fig. 6 | Clade 2.3.4.4b H5 HA mRNA-LNP vaccine elicits antibodies in ferrets.** Four ferrets per group were immunized with 60 µg A/Astrakhan/3212/2020 HA mRNA-LNP (H5 mRNA-LNP) or Luciferase mRNA-LNP (control), followed by a 60 µg boost 28 days later. Ferret serum samples were collected before vaccination and 28 days after each vaccination. **A, B** A/Astrakhan/3212/2020 and (**C, D**) A/pheasant/New York/22-009066-001/2022 serum IgG binding and 50% Foci reduction neutralization test (FRNT50) titers were quantified. Reciprocal dilutions of serum required to inhibit 50% virus infection are shown in panels (**B**) and (**D**). Serum IgG titers to the 'headless' H5 stalk protein were quantified after (**E**) prime and (**F**) boost.

Data are shown as geometric means ± 95% confidence intervals and values were log-transformed before statistical analysis. Data in A-D were analyzed by mixed-model ANOVA with Greenhouse-Geisser correction and Sidak's multiple comparisons test to compare differences with luciferase (control) mRNA immunization. Data in E-F were analyzed using a two-tailed Mann-Whitney test. **A** Day 28 *$P = 0.0016$, day 56 *$P = 0.0001$; (**B**) *$P = 0.0008$; (**C**) Day 28 *$P < 0.0001$, day 56 *$P = 0.002$; (**D**) Day 28 *$P < 0.0001$, day 56 *$P = 0.0005$; (**E**) *$P = 0.0286$; (**F**) *$P = 0.0286$ . Source data are provided as a Source Data file.

PBS, and 25 µl was injected in each hind leg. Blood samples were obtained by submandibular bleeding 28, 100, and 365 days after vaccination, and sera were isolated by centrifugation using Z-Gel tubes (Sarsedt). For T cell experiments, spleens were harvested 10 days after vaccination.

**H1N1 prior exposure experiments in mice**

For some experiments, female C57BL/6 mice aged 6-8 weeks were intranasally infected with 1000 TCID$_{50}$ of A/California/07/2009 diluted in 20 µL of PBS. For these experiments, the A/California/07/2009 virus possessed a D225G HA mutation that increases infectivity in mice

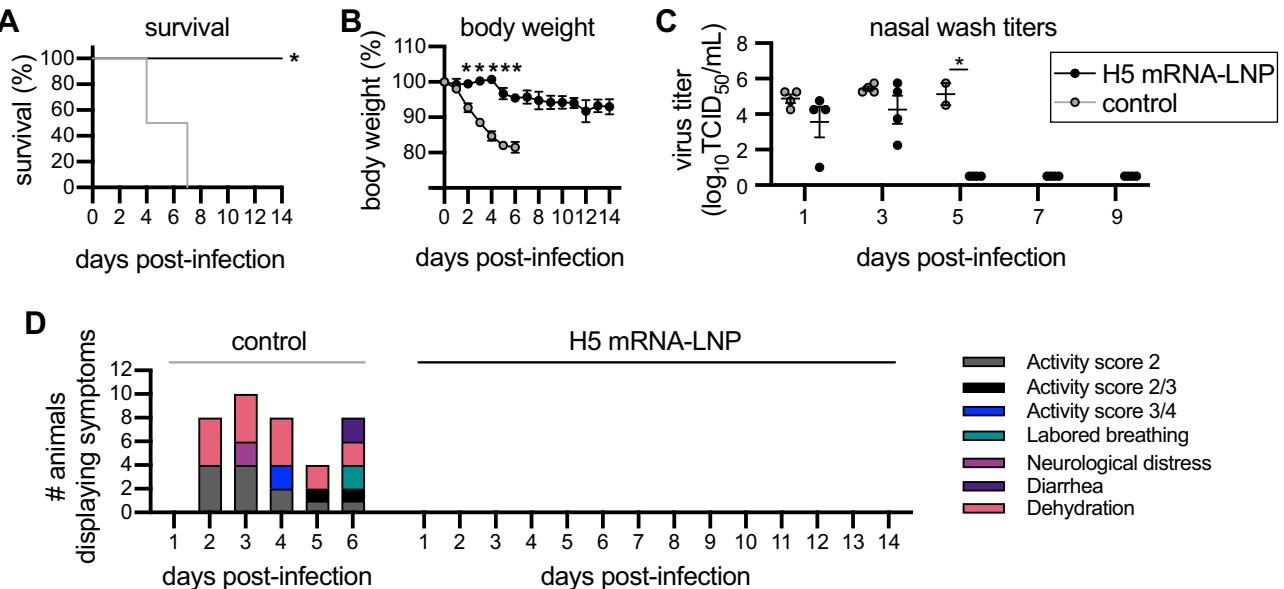

**Fig. 7 | H5 HA mRNA-LNP vaccine protects ferrets from clade 2.3.4.4b H5 virus infection.** Four ferrets per group were immunized with 60 µg A/Astrakhan/3212/2020 HA mRNA-LNP (H5 mRNA-LNP) or Luciferase mRNA-LNP (control), followed by a 60 µg boost 28 days later. Ferrets were challenged i.n. with A/bald eagle/Florida/ W22-134-OP/2022 28 days after the second vaccination and then monitored for 14 days after infection. **A** Survival, (**B**) body weight, (**C**) 50% tissue culture infectious dose (TCID50) virus titers in nasal wash samples and (**D**) clinical scores are reported after infection. Data in (**A**) were analyzed using a log rank test. Data in (**B**, **C**) are shown as means ± SEMs. Data in (**B**, **C**) were analyzed by multiple unpaired two-sided *t* tests with Gaussian distribution to compare differences with luciferase (control) mRNA immunization. **A** *P = 0.0009; (**B**) Day 2 *P = 0.005166, day 3 *P = 0.000095, day 4 *P = 0.000087, day 5 *P = 0.003561 and day 6 *P = 0.001509; (**C**) *P = 0.000269. Source data are provided as a Source Data file.

as previously reported[35]. These mice were then immunized intramuscularly with either 1 µg of H5 mRNA-LNP vaccine or 50 hemagglutinating units (HAU) of an inactivated H5 vaccine 90 days after A/California/08/2009 infection. We also immunized mice without prior H1N1 exposure intramuscularly with either 1 µg of H5 mRNA-LNP vaccine or 50 HAU of an inactivated H5 vaccine. Serum was obtained 30 days after vaccination. The inactivated vaccine was produced by β-propiolactone (BPL) inactivation of concentrated egg-grown CBER-RG8A (contains HA and NA from A/Astrakhan/3212/2020 (H5N8)) virus at the St. Jude Children's Research Hospital. Any mice with prior exposure to A/California/07/2009 that did not develop binding and neutralizing antibodies to A/California/07/2009 were determined to not have been properly infected and were removed from our analyses.

### Ferret experiments
Ferret studies were approved by the St. Jude Children's Research Hospital Institutional Animal Care and Use Committee (IACUC, protocol number 428) in accordance with the guidelines established by the Institute of Laboratory Animal Resources, approved by the Governing Board of the US National Research Council, and carried out by trained personnel working in a United States Department of Agriculture (USDA)-inspected Animal Biosafety Level 3+ animal facility in accordance with all regulations established by the Division of Agricultural Select Agents and Toxins (DASAT) at the USDA Animal and Plant Health Inspection Service (APHIS), as governed by the United States Federal Select Agent Program (FSAP) regulations (7 CFR Part 331, 9 CFR Part 121.3, 42 CFR Part 73.3). This study utilized USDA-classified select agents and A(H5N1) viruses used herein are subject to the guidelines of, and compliance with, requirements discussed in Title 9 (CFR Parts 121 [Possession, Use, and Transfer of Select Agent Toxins] and 122 [Importation and Transportation of Controlled Organisms and Vectors]). Four-to-six-month-old influenza-seronegative male ferrets (Triple F Farms, Sayre, PA, USA) were primed with 60 µg of mRNA-LNP

vaccine encoding H5 (*n* = 4 animals) or an irrelevant protein (Luciferase, *n* = 4 animals) and then boosted 28 days later with the same vaccine. Blood samples were collected 28 days after each vaccine dose. Animals were lightly anesthetized with isoflurane 28 days after the second vaccination and inoculated intranasally with $10^6$ EID50 units of A/bald eagle/Florida/ W22-134-OP/2022 (H5N1) diluted in 1.0 mL of PBS. All animals were monitored daily for clinical signs of infection. Characteristics monitored included body temperature, weight loss, relative inactivity indices, ataxia, respiratory symptoms, stool consistency, and neuropathologic signs. Animals reaching the humane endpoint, according to an IACUC-approved clinical scoring system, were euthanized. Nasal washes were collected from all surviving ferrets at 1, 3, 5, 7, and 9 days post-infection (dpi). Ketamine was used to induce sneezing. Viral titers were determined in MDCK cells by TCID50 assay.

### Recombinant influenza virus HA proteins
Recombinant HA (rHA) proteins were generated using pCMV-Sport6 vectors encoding full-length, codon-optimized HA sequences from A/Astrakhan/3212/2020 (EPI ID 1038924), A/pheasant/New York/22-009066-001/2022 (EPI ID 11971502), A/red fox/England/AVP-M1-21-01/2020 (EPI ID 2081527), A/Vietnam/1203/2004 (EPI ID 116507), A/Hubei/1/2010 (Genbank CY098760.1), A/Indonesia/5/2005 (Genbank CY116648.1), or the "headless" HA stalk of A/Vietnam/1203/2004. For each of these proteins, the HA transmembrane domain was replaced with the Foldon T4 trimerization domain of T4 fibritin, an Avitag site-specific biotinylation sequence, and a hexahistidine tag as described previously[36]. To produce the recombinant proteins, rHA plasmid and a plasmid encoding neuraminidase (NA) from A/Puerto Rico/8/1934 were co-transfected into 293 F suspension cells (Thermo Fisher) using 293Fectin (Thermo Fisher). Supernatants were collected 4 days later for protein purification by Ni-NTA affinity chromatography (Qiagen).

**Table 1 | Information on the antibodies that were used for T cell assays**

| Antibody | Catalog # | Clone # | Manufacturer | Fluorophore | Dilution |
|---|---|---|---|---|---|
| IFNg | 11-7311-41 | XMG1.2 | Ebioscience | FITC | 1:400 |
| Foxp3 | 35-5773-82 | FJK-16s | Ebioscience | PECy5.5 | 1:200 |
| CXCR5-biotin | 13-7185-82 | SPRCL5 | Ebioscience | n/a | 1:50 |
| Streptavidin | 405237 | n/a | Biolegend | AF647 | 1:500 |
| CD90.2 | 140324 | 53-2.1 | Biolegend | AF700 | 1:400 |
| Ghostdye Red 780 | 13-0865-T100 | n/a | Tonbo | n/a | 1:500 |
| KLRG1 | 740279 | 2F1 | BD | BUV395 | 1:200 |
| CD4 | 612952 | GK1.5 | BD | BUV496 | 1:400 |
| CD8a | 752637 | 5H10-1 | BD | BUV563 | 1:400 |
| CD122 | 741537 | 5H4 | BD | BUV661 | 1:200 |
| CD69 | 612793 | H1.2F3 | BD | BUV737 | 1:200 |
| CD11a | 741919 | 2D7 | BD | BUV805 | 1:500 |
| TNFa | 506328 | MP6-XT22 | Biolegend | BV421 | 1:300 |
| PD-1 | 135220 | 29 F.1A12 | Biolegend | BV605 | 1:200 |
| CD19 | 115541 | 6D5 | Biolegend | BV650 | 1:400 |
| CD62L | 104445 | MEL-14 | Biolegend | BV711 | 1:400 |
| CD44 | 103059 | IM7 | Biolegend | BV785 | 1:400 |
| IL-4 | 504104 | 11B11 | Biolegend | PE | 1:200 |
| Bcl-6 | 562401 | K112-91 | BD | PE-CF594 | 1:50 |
| T-bet | 15-5825-82 | 4B10 | Ebioscience | PECy5 | 1:200 |
| CD107a | 121620 | 1D4B | Biolegend | PECy7 | 1:400 |

### Enzyme-linked immunosorbent assays (ELISAs)

ELISAs were performed using 96 well plates (Immulon) coated with 2 µg/mL rHA overnight at 4 °C. Blocking and dilution buffer consisted of 1xPBS with 0.1% Tween 20, 0.5% milk, and 3% goat serum. ELISA plates were blocked for 1 h at room temperature. Heat-inactivated serum samples were serially diluted two-fold or three-fold in round bottom 96 well plates and were then added to ELISA plates and incubated for 2 h at room temperature. As a standard control to establish relative serum IgG titers, each ELISA plate included a serial dilution of the CR9114 human monoclonal antibody. Horseradish-peroxidase-conjugated goat anti-mouse IgG (Jackson, 115-035-003; diluted 1:1000), anti-ferret IgG (Abcam, ab112770; diluted 1:2500), or anti-human IgG (Jackson, 109-036-098; diluted 1:5000) were added to ELISA plates and incubated for 1 h. Plates were developed by adding SureBlue TMB substrate (SeraCare) and quenching the development reaction after 5 min using 250 mM hydrochloric acid. Plates were read using a SpectraMax 190 microplate reader (Molecular Devices) at an optical density (OD) of 450 nm.

### H5 in vitro neutralization assays

Neutralization assays for H5 viruses were performed using viruses expressing GFP in place of the PB1 gene that can only replicate in cells expressing PB1, as previously described[37]. Viruses were generated by reverse genetics using the plasmid pHH-PB1flank-GFP that encodes GFP in the open reading frame of the viral polymerase PB1 gene. To generate the viruses, a coculture of 293T-CMV-PB1 and MDCK-SIAT1-TMPRSS2-PB1 cells were transfected in serum-free DMEM with 5 bidirectional reverse genetics plasmids encoding PB2, PA, NP, M, and NS from A/Puerto Rico/8/1934 along with pHH-PB1flank-GFP and 2 reverse genetics plasmids encoding the HA and NA segments from clade 2.3.4.4b H5 viruses (A/Astrakhan/3212/2020, A/pheasant/New York/22-009066-001/2022, and A/red_fox/England/AVP-M1-21-01/ 2020). As an additional biosafety precaution, the multibasic cleavage

site KRRKR was edited to a monobasic cleavage site R in the reverse genetics HA plasmids. At 20 h post-transfection, media was changed to neutralization assay media (NAM) consisting of Medium 199 (Thermo Fisher) with 0.01% heat-inactivated fetal bovine serum, 0.3% bovine serum albumin, 100 µg of penicillin/ml, 100 µg of streptomycin/ml, 100 µg of calcium chloride/ml, and 25 mM HEPES. Supernatants were collected and clarified by centrifugation at 72 h post transfection and clarified supernatants were expanded on subconfluent MDCK-SIAT1-TMPRSS2-PB1 cells for an additional 72 h. Expansion supernatants were collected, clarified, and titrated on MDCK-SIAT1-TMPRSS2-PB1 cells using an EnVision microplate reader at an excitation wavelength of 485 nm and an emission wavelength of 530 nm. Neutralization assays were performed using serum treated with receptor-destroying enzyme (RDE; Denka Seiken) followed by heat-inactivation. Sera were serially diluted two-fold with NAM in flat bottom 96 well plates before an equal volume of virus was added. Serum-virus mixtures were incubated at 37 °C in 5% $CO_2$ for 1 h before $2 \times 10^5$ MDCK-SIAT1-TMPRSS2-PB1 cells were added to each well. At 40 h post-infection, cells were fixed in 4% paraformaldehyde and GFP fluorescence intensities were measured on an EnVision plate reader as described above. 50% neutralization titers are reported as the highest reciprocal serum dilution that decreased GFP levels by 50% or greater in relation to the virus only control wells from each plate, as described previously[24]. Serum samples that were unable to reduce GFP levels by 50% at a 1:40 dilution were given a 50% neutralization titer of 20.

### H1 in vitro neutralization assays

A/California/07/2009 H1N1 virus engineered to possess a D225G HA mutation was created by transfecting a coculture of 293 T and MDCK-SIAT1 cells in serum-free DMEM with bidirectional reverse genetics plasmids encoding PB1, PB2, PA, NP, M and NS from A/Puerto Rico/8/1934 along with the reverse genetics plasmids encoding the HA and NA segments from A/California/07/2009. At 20 h post-transfection, the media was changed to NAM, and at 72 h post-transfection, supernatants were harvested and clarified by centrifugation. Supernatants were then injected into 10 day old fertilized chicken eggs. Eggs were incubated at 37 °C for 48 hrs and then the virus was collected. Virus was titrated on MDCK-SIAT1 cells to determine the dilution that results in 300 focus-forming units per well. The FRNT assay was performed as previously described[38]. Briefly, serum samples were treated with RDE and heat-inactivated. MDCK-SIAT1 cells were seeded the day before the assay at $2.5 \times 10^4$ cells per well in a flat-bottom 96-well plate. The day of the assay, sera were serially diluted two-fold with serum-free MEM (SF-MEM), followed by the addition of an equal volume of virus, and then incubated for 1 h at 37 °C. Cells were washed twice with SF-MEM and then the serum-virus mixture was added to the cells and incubated for 1 h at 37 °C. Following incubation, the serum-virus mixture was removed from the cells, and cells were washed twice and overlayed with 50% SF-MEM, 1.25% avicel, 5 mM HEPES, and 50 µg/mL Gentamycin-Sulfate. 18 h post-infection, cells were fixed in 4% paraformaldehyde and permeabilized with 0.5% Triton-X 100 in 1X PBS. Plates were then blocked for 1 h with 5% milk in 1X PBS. Following blocking, plates were stained with mouse anti-influenza A NP (IC5-1B7, produced in-house) at a 1:20,000 dilution in 5% milk for 1 h, followed by staining with rat anti-mouse HRP (Southern Biotech: 1170-05) at a 1:30,000 dilution in 5% milk for 1 h. Plates were developed using TrueBlue TMB substrate and read on an ELISpot reader. The FRNT titers for 50% virus inhibition are reported and samples that were unable to reduce 50% at a 1:20 dilution are reported as having a titer of 10.

### HA multiplex binding assays

Recombinant HA proteins were coupled to MagPlex beads (Luminex) following the manufacturer's instructions at ratios of 0.1 nmol antigen/

$1 \times 10^6$ beads. Beads coupled with HA proteins and blank beads to act as a negative control were added to 96-well black, clear-bottom plates at a concentration of 2500 beads/well (per bead, i.e. for this assay with 7 bead regions, a total of 17,500 beads were added to each well). RDE-treated sera were diluted 1:400 in PBS-TBN (1X PBS, 0.1% bovine serum albumin, 0.02% tween20, and 0.05% sodium azide), added to the wells with the beads, and incubated on a plate shaker for 1 h at 600 rpm. Plates were washed twice with PBS-TBN and then r-phycoerythrin affinipure goat anti-mouse IgG (subclasses 1 + 2a + 2b + 3) Fcγ fragment specific secondary antibody (Jackson Immunoresearch: 115-115-164) diluted to 4 μg/mL in PBS-TBN was added to each well. After a 30 min incubation on the plate shaker, plates were washed twice with PBS-TBN and 100 μL of PBS-TBN was added to each well. Plates were then read on an Intelliflex (Luminex), and the median mean fluorescent intensity (MFI) was reported.

## T cell assays

A pool of 139 overlapping peptides (15mers, overlapping adjacent peptides by 4 amino acids) spanning the HA sequence of A/Astrakhan/3212/2020 (H5) were prepared (≥70% purity, GenScript) and pooled for in vitro restimulation assays. Mice were vaccinated with H5 mRNA-LNP or control mRNA-LNP and spleens were isolated 10 days later. Spleens were processed into single cell suspensions, plated ($5 \times 10^6$ cells/well), and cultured in the presence or absence of the peptide pool (10 μg/mL) for 6 hr at 37 °C. Protein Transport Inhibitor Cocktail (Ebioscience: 00-4980-03) was added to cultures after 3 hr for intracellular staining of cytokines and markers of degranulation. Following incubation cultures were harvested and stained for flow cytometry analysis performed on BD FACSymphony A3. Table 1 summarizes information on the antibodies that were used to assess T cell responses to the peptide pool. Analysis was performed in FlowJo v10.10.0 with automatic quality control performed by the flowAI plugin[39]. All gating was done from FlowAIGoodEvents subset of events following quality control. Doublet discrimination and gating out of dead cells was always performed primary to analysis of responding T cells.

## Quantifying nasal wash viral titers

Madin–Darby canine kidney (MDCK) cells (ATCC CCL-34) were cultured in Modified Eagle's Medium (MEM) (CellGro) supplemented with 5% fetal bovine serum (FBS) (HyClone), 1 mM L-glutamine, and 1× penicillin/streptomycin/amphotericin B (Gibco) and NaHCO₃ (1.5 g/L). Cells were maintained at 37 °C in 5% $CO_2$. Infectious viral titers of ferret nasal washes were determined by performing 10-fold dilutions on all nasal wash samples. MDCK monolayers were inoculated with 100 μL of diluted sample and then incubated at 37 °C for 72 h. At 72 HPI 50 μL of cell supernatant was mixed with 50 μL of 0.5% chicken red blood cells (CRBC) to measure hemagglutinin agglutination. The endpoint was defined as the highest virus dilution resulting in CRBC agglutination. The 50% tissue culture infectious dose ($TCID_{50}$) titer was determined by the Reed and Muench method[40] and the lower limit of virus detection was 1.0 $\log_{10}$ $TCID_{50}$/mL.

## Statistical Analysis

Data were analyzed using GraphPad Prism version 9.3.0.

## Reporting summary

Further information on research design is available in the Nature Portfolio Reporting Summary linked to this article.

## Data availability

Datasets generated and/or analyzed during the current study are included in the paper or are appended as supplementary data. Source data are provided with this paper.

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

## Acknowledgements

This project has been funded in part with Federal funds from the National Institute of Allergy and Infectious Diseases, National Institutes of Health, Department of Health and Human Services, under Contract Nos. 75N93021C00015 (S.E.H.) and 75N93021C00016 (R.J.W.), and grant numbers R01AI08686 (S.E.H.) and R01AI126899 (C.A.H.). Funding was also received from the Commonwealth of Pennsylvania (S.E.H. and C.A.H.) and the Penn Institute for Infectious and Zoonotic Diseases (C.A.H.). S.E.H. holds an Investigators in the Pathogenesis of Infectious Disease Awards from the Burroughs Wellcome Fund.

## Author contributions

C.F. and S.E.H. designed the experiments, analyzed, and interpreted data and wrote the manuscript. C.F., G.S. and N.Y. completed mouse experiments and serological experiments. L.K., J.D., J.C., T.J., C.P. and J.F. completed ferret experiments. A.R. prepared an inactivated vaccine for mouse experiments. A.T.P. completed T cell experiments. M.-G. A., S.H.Y.F. and D.W. designed and produced mRNA-LNP and provided technical advice. C.A.H. supervised T cell experiments and R.J.W. supervised ferret experiments. S.E.H. supervised all other activities.

## Competing interests

S.E.H. and D.W. are co-inventors on patents that describe the use of nucleoside-modified mRNA as a platform to deliver therapeutic proteins and as a vaccine platform. S.E.H. reports receiving consulting fees from Sanofi, Pfizer, Lumen, Novavax, and Merck. S.H.Y.F. is an employee of Acuitas Therapeutics, a company focused on the development of lipid nanoparticulate nucleic acid delivery systems for therapeutic applications. The authors declare no other competing interests.
