## [Peer Review File · Nature Communications]

REVIEWER COMMENTS

Reviewer #1 (Remarks to the Author):

In this short report, the authors develop an LNP formulated mRNA vaccine against current circulating H5. They tested it for immunogenicity in mice and protection in ferrets. The work is clearly presented.

Suggest some more discussion to explain how this moves the field forwards? mRNA vaccines for influenza have been shown to be protective and it would be extremely surprising if it didn't work for this antigen.

Minor comments:

1. In panel 2G can you separate the points so you can see all 4 animals.

Reviewer #2 (Remarks to the Author):

Furey et al develop an mRNA-LNP vaccine against the 2.3.4.4b clade of H5 avian IAV that has been causing significant morbidity and mortality across animal globally. The authors test their vaccine in mice showing strong immunogenicity for antibody and T cell responses and then demonstrate protective capacity of their vaccine in a ferret challenge. The manuscript is overall well written and highly relevant to the field. I have the following minor suggestions that I believe would strengthen the manuscript prior to publication:

- 1) The authors should acknowledge the dose dependence of the antibody responses observed, especially for the heterologous antigens tested.

- 2) Can the authors comment on the antigenic diversity of 2.4.4.b H5 viruses? (if such data are available). Although the vaccine is evidently very potent against severe disease and death, the

authors should acknowledge that the protection provided in ferrets (against a heterologous strain) is non-sterilising and, depending on the antigenic diversity of 2.3.4.4b viruses, this will likely apply to humans or animals. This does not detract from the relevance or importance of the study.

3) The authors should clarify the dose used for the T cell studies.

4) The authors should provide additional information for the T cell studies.

a. What vaccine dose was used? From figure legend 1G-H it seems like 1ug was used, please clarify that in the methods section too.

b. Additional information of the staining is required – what markers/antibodies/clones were used and provide a gating strategy with representative flow cytometry plots.

Reviewer #3 (Remarks to the Author):

Authors describe in a very brief report the immunogenicity of mRNA vaccine expressing HA of A/Astrakhan/3212/2020 H5N8 virus in naïve ferrets. The authors found that animals vaccinated with two doses of this vaccine are protected against a lethal challenge with H5N1 virus representative of viruses circulating in USA in 2022 (i.e. A/bald eagle/Florida/W22-134-OP/2022). Both the vaccine strain and the challenge virus belong to clade 2.3.4.4b, which indicates that they have very similar HA sequences. The paper is well written, and the experiments are described properly but the report is too brief to be published in a high-impact journal.

Although there is an increased number of outbreaks with highly pathogenic avian influenza H5Nx in birds and mammals in the last years across the globe and public health leaders are concerned that these viruses may acquire the ability to infect humans, I found that this report is not sufficient to evaluate the mRNA vaccine platform for emerging influenza viruses. In contrast with the COVID-19 pandemic, there is a decades-old program led by WHO for surveillance of influenza viruses with pandemic potential and rapid vaccine development of H5Nx vaccines using inactivated or live attenuated viruses. Using influenza reverse genetics, an influenza candidate vaccine virus (including live-attenuated virus) can be rescued in a few weeks and the vaccine virus can propagate safely in eggs or mammalian cells for rapid vaccine production across the globe. Although we need alternative approaches for rapid influenza vaccine development, the main challenges are to develop vaccines able to confer protection in humans, which have pre-existing immunity from infection with seasonal influenza, not only against the emerging strain but also against novel variants, which may emerge

after the virus will start to spread in humans, and to slow the virus transmission by eliciting mucosal immunity.

These groups are well prepared to publish a thorough study addressing some of these issues:

-to compare the breadth, depth and durability elicited by mRNA, inactivated and live-attenuated H5Nx virus vaccines

-to study the breadth, depth and durability of immune responses elicited by influenza vaccines not only in naïve animals but also in animals pre-exposed to seasonal influenza.

These are some of the main questions, which need to be addressed now if we want to develop an efficient influenza pandemic response:

Which influenza vaccine can elicit durable protection against H5NX viruses belonging to different genetic clades comprising viruses from human outbreaks (e.g., clade 1.1, 2.1.3.x, 2.3.2.1x, 2.3.4.4, etc)?

Which vaccine (inactivated virus, live-attenuated virus, protein-based or mRNA) can elicit not only neutralizing antibodies against the HA head, the most variable region of HA, but also against the more conserved HA stalk region. How durable are these responses?

Which vaccine can elicit broad, potent, and durable protection not only in naïve animals but also in animals exposed to seasonal influenza?

Which H5 vaccine can confer not only protection against severe disease and death but slow virus transmission by eliciting mucosal responses?

Point by point for NCOMMS-23-19576-T

Below is a point-by-point response (*our response in italic*) that includes a detailed description of how we responded to the reviewers.

Overall summary: We transformed our manuscript from a Brief Report → a full length article. We complete several new experiments suggested by the reviewer, resulting in 5 (!!!) new figures, as outlined below. This was a massive effort and we hope that our manuscript is now suitable for publication in Nature Communications.

REVIEWER COMMENTS

Reviewer #1 (Remarks to the Author):

In this short report, the authors develop an LNP formulated mRNA vaccine against current circulating H5. They tested it for immunogenicity in mice and protection in ferrets. The work is clearly presented.

Suggest some more discussion to explain how this moves the field forwards? mRNA vaccines for influenza have been shown to be protective and it would be extremely surprising if it didn't work for this antigen.

Response: We revised our manuscript to be a full-length paper rather than a short report. In our revision, we have a new Discussion section that describes how our study moves the field forward and why mRNA-LNP vaccines might be useful during an influenza pandemic.

Minor comments:

1. In panel 2G can you separate the points so you can see all 4 animals.

Response: We edited this figure panel as suggested.

Reviewer #2 (Remarks to the Author):

Furey et al develop an mRNA-LNP vaccine against the 2.3.4.4b clade of H5 avian IAV that has been causing significant morbidity and mortality across animal globally. The authors test their vaccine in mice showing strong immunogenicity for antibody and T cell responses and then demonstrate protective capacity of their vaccine in a ferret challenge. The manuscript is overall well written and highly relevant to the field. I have the following minor suggestions that I believe would strengthen the manuscript prior to publication:

1) The authors should acknowledge the dose dependence of the antibody responses observed, especially for the heterologous antigens tested.

Response: We edited the text as suggested.

2) Can the authors comment on the antigenic diversity of 2.4.4.b H5 viruses? (if such data are available). Although the vaccine is evidently very potent against severe disease and death, the authors should acknowledge that the protection provided in ferrets (against a heterologous

strain) is non-sterilising and, depending on the antigenic diversity of 2.3.4.4b viruses, this will likely apply to humans or animals. This does not detract from the relevance or importance of the study.

Response: We edited the text as suggested and describe how the HA of the challenge strain has 4 amino acid differences compared to the HA of the vaccine strain.

3) The authors should clarify the dose used for the T cell studies.

Response: We clarified as suggested.

4) The authors should provide additional information for the T cell studies.

a. What vaccine dose was used? From figure legend 1G-H it seems like 1ug was used, please clarify that in the methods section too.

Response: We clarified as suggested; 1ug doses were used for the T cell studies.

b. Additional information of the staining is required – what markers/antibodies/clones were used and provide a gating strategy with representative flow cytometry plots.

Response: We added this information to the Methods as requested. In our revision, we added representative flow plots in our new Figure 4A-B.

Reviewer #3 (Remarks to the Author):

Authors describe in a very brief report the immunogenicity of mRNA vaccine expressing HA of A/Astrakhan/3212/2020 H5N8 virus in naïve ferrets. The authors found that animals vaccinated with two doses of this vaccine are protected against a lethal challenge with H5N1 virus representative of viruses circulating in USA in 2022 (i.e. A/bald eagle/Florida/W22-134-OP/2022). Both the vaccine strain and the challenge virus belong to clade 2.3.4.4b, which indicates that they have very similar HA sequences. The paper is well written, and the experiments are described properly but the report is too brief to be published in a high-impact journal.

Although there is an increased number of outbreaks with highly pathogenic avian influenza H5Nx in birds and mammals in the last years across the globe and public health leaders are concerned that these viruses may acquire the ability to infect humans, I found that this report is not sufficient to evaluate the mRNA vaccine platform for emerging influenza viruses. In contrast with the COVID-19 pandemic, there is a decades-old program led by WHO for surveillance of influenza viruses with pandemic potential and rapid vaccine development of H5Nx vaccines using inactivated or live attenuated viruses. Using influenza reverse genetics, an influenza candidate vaccine virus (including live-attenuated virus) can be rescued in a few weeks and the vaccine virus can propagate safely in eggs or mammalian cells for rapid vaccine production across the globe. Although we need alternative approaches for rapid influenza vaccine development, the main challenges are to develop vaccines able to confer protection in humans, which have pre-existing immunity from infection with seasonal influenza, not only against the emerging strain but also against novel variants, which may emerge after the virus will start to spread in humans, and to slow the virus transmission by eliciting mucosal immunity.

These groups are well prepared to publish a thorough study addressing some of these issues: -to compare the breadth, depth and durability elicited by mRNA, inactivated and live-attenuated H5Nx virus vaccines

-to study the breadth, depth and durability of immune responses elicited by influenza vaccines not only in naïve animals but also in animals pre-exposed to seasonal influenza. These are some of the main questions, which need to be addressed now if we want to develop an efficient influenza pandemic response:

Which influenza vaccine can elicit durable protection against H5NX viruses belonging to different genetic clades comprising viruses from human outbreaks (e.g., clade 1.1, 21.3.x, 2.3.2.1x, 2.3.4.4, etc)?

Response: *We completed new experiments to address the Reviewer's comment about durability. In our revision, we present new antibody data using samples collected from mice 1 year (!!!) after vaccination (new Figure 1A, 1B, 1C). We found that antibodies elicited by mRNA-LNP vaccines decline over time but remain at high levels 1 year after vaccination.*

In our revision, we added new data where we quantified HA stalk-reactive antibodies elicited in mice (new Figure 1B) and ferrets (new Figure 6 E-F) following mRNA-LNP vaccination. We found that the mRNA-LNP vaccine elicited high levels of HA stalk-reactive antibodies that persisted for one year.

In our revision, we also completed new multianalyte bead-based experiments to measure antibody binding to clade 2.3.4.4b viruses, a clade 1 virus (A/Vietnam/1203/2004), a clade 2.3.2.1a virus (A/Hubei/1/2010), and a clade 2.1.3.2 virus (A/Indonesia/5/2005) (new Figure 3A-F). We found that antibodies elicited by the vaccine bound to clade 1, clade 2.3.2.1a, and clade 2.1.3.2 viruses; albeit at lower levels compared to binding to clade 2.3.4.4b viruses (as expected).

Which vaccine (inactivated virus, live-attenuated virus, protein-based or mRNA) can elicit not only neutralizing antibodies against the HA head, the most variable region of HA, but also against the more conserved HA stalk region. How durable are these responses?

Response: *In our revision, we added new data where we quantified HA stalk-reactive antibodies elicited in mice 1 year after vaccination (new Figure 1B) and ferrets 28 days after vaccination (new Figure 6 E-F). We also completed a new set of experiments directly comparing mRNA-LNP and inactivated vaccines in mice with and without prior H1N1 exposures (new Figure 5A-E). We found that both vaccine types elicit high levels of antibodies in mice with and without prior H1N1 exposures.*

Which vaccine can elicit broad, potent, and durable protection not only in naïve animals but also in animals exposed to seasonal influenza?

Response: *As noted above, we completed a new set of experiments directly comparing mRNA-LNP and inactivated vaccines in mice with and without prior H1N1 exposures (new Figure 5A-E). We found that both vaccine types elicit high levels of antibodies in mice with and without prior H1N1 exposures.*

Which H5 vaccine can confer not only protection against severe disease and death but slow virus transmission by eliciting mucosal responses?

Response: *We think that this is interesting but would be more appropriate for a separate study. Transmission studies would need to be completed in ferrets and we do not have sufficient funding to complete these types of experiments.*

REVIEWERS' COMMENTS

Reviewer #1 (Remarks to the Author):

thank you for adding the extra work.

The discussion needs to be more focused on comparison to other studies and demonstrating the conceptual advances than summarising the data presented.

Reviewer #2 (Remarks to the Author):

The authors have addressed my comments

Reviewer #3 (Remarks to the Author):

The authors revised the manuscript describing the characterization of the H5 mRNA vaccine by performing several new experiments. However, I am disappointed about their decisions to measure the breadth of serum antibodies by a binding assay instead of a neutralization assay and to study vaccine immunogenicity in mice previously infected with H1N1 instead of doing this experiment in ferrets.

For the neutralization assays, authors performed the experiments using an H5N1 virus in which the PB1 gene was replaced with a fluorescent reporter. This virus is safe to use in BSL2 laboratory because it can replicate only in cells expressing PB1. This approach greatly facilitates the characterization of neutralizing antibodies. Why did not the authors prepare a panel of H5N1 viruses and measure the breadth and depth of neutralizing serum antibodies instead of measuring the binding to different H5 HA proteins?

The authors compared the immunogenicity of mRNA and inactivated H5 virus vaccines in mice pre-infected with H1N1. I am disappointed by their decision to do this experiment in mice instead of ferrets, the gold-standard animal model for influenza studies. What is the significance of this result for human vaccine studies? The authors are leaders in the field, and I hope they will not set an example with this experiment, which has minimal value for human influenza vaccine studies. At least, the authors should discuss the limitations of their experiment performed in mice pre-exposed to H1N1.

Overall, this paper describes well that the H5 mRNA vaccine is immunogenic in mice and ferrets and it can protect against severe disease in ferrets challenged with an H5N1 strain genetically similar to the vaccine strain. I hoped the authors would raise the bar in how we characterize influenza vaccines in animal studies, but they did not.

Point by point for NCOMMS-23-19576A

Below is a point-by-point response (*our response in italic*) that includes a detailed description of how we responded to the reviewers.

REVIEWER COMMENTS

Reviewer #1 (Remarks to the Author):

thank you for adding the extra work.

The discussion needs to be more focused on comparison to other studies and demonstrating the conceptual advances than summarising the data presented.

Response: We added references and several new points to the Discussion.

Reviewer #2 (Remarks to the Author):

The authors have addressed my comments

Reviewer #3 (Remarks to the Author):

The authors revised the manuscript describing the characterization of the H5 mRNA vaccine by performing several new experiments. However, I am disappointed about their decisions to measure the breadth of serum antibodies by a binding assay instead of a neutralization assay and to study vaccine immunogenicity in mice previously infected with H1N1 instead of doing this experiment in ferrets.

For the neutralization assays, authors performed the experiments using an H5N1 virus in which the PB1 gene was replaced with a fluorescent reporter. This virus is safe to use in BSL2 laboratory because it can replicate only in cells expressing PB1. This approach greatly facilitates the characterization of neutralizing antibodies. Why did not the authors prepare a panel of H5N1 viruses and measure the breadth and depth of neutralizing serum antibodies instead of measuring the binding to different H5 HA proteins?

Response: We found that most of the cross-reactive H5 antibodies target the H5 stalk and therefore we do not expect that these antibodies will neutralize the distinct clades of H5 virus (since stalk antibodies are typically poor at neutralizing virus). We described this in our manuscript.

The authors compared the immunogenicity of mRNA and inactivated H5 virus vaccines in mice pre-infected with H1N1. I am disappointed by their decision to do this experiment in mice instead of ferrets, the gold-standard animal model for influenza studies. What is the significance of this result for human vaccine studies? The authors are leaders in the field, and I hope they will not set an example with this experiment, which has minimal value for human influenza vaccine studies. At least, the authors should discuss the limitations of their experiment performed in mice pre-exposed to H1N1.

Response: *Who is to say that prior exposure experiments in ferrets are more relevant than prior exposure experiments in mice? It's a tricky business trying to mimic complex human influenza virus exposure histories in animals with short lifespans. We have been completing these kinds of experiments for years, and we do not think there will be fundamental differences comparing mice versus ferrets for prior-exposure experiments. We have previously published experiments demonstrating the utility of the mouse prior exposure model: Arevalo et al. Science 2022.*

Overall, this paper describes well that the H5 mRNA vaccine is immunogenic in mice and ferrets and it can protect against severe disease in ferrets challenged with an H5N1 strain genetically similar to the vaccine strain. I hoped the authors would raise the bar in how we characterize influenza vaccines in animal studies, but they did not.

Response: *We think this study is an important contribution and appreciate that you reviewed the manuscript critically.*